# Stereoselective C–O silylation and stannylation of alkenyl acetates

Ying Hu[1], Jiali Peng[2,3], Binjing Hu[1], Jixin Wang[1], Jing Jing[1], Jie Lin[1], Xingchen Liu[1], Xiaotian Qi [2] & Jie Li [1,4] ✉

Facile formation of carbon-heteroatom bonds is a long-standing objective in synthetic organic chemistry. However, direct cross-coupling with readily accessible alkenyl acetates via inert C–O bond-cleavage for the carbon-heteroatom bond construction remains challenging. Here we report a practical preparation of stereoselective tri- and tetrasubstituted alkenyl silanes and stannanes by performing cobalt-catalyzed C–O silylation and stannylation of alkenyl acetates using silylzinc pivalate and stannylzinc chloride as the nucleophiles. This protocol features a complete control of chemoselectivity, stereoselectivity, as well as excellent functional group compatibility. The resulting alkenyl silanes and stannanes show high reactivities in arylation and alkenylation by Hiyama and Stille reactions. The synthetic utility is further illustrated by the facile late-stage modifications of natural products and drug-like molecules. Mechanistic studies suggest that the reaction might involve a chelation-assisted oxidative insertion of cobalt species to C–O bond. We anticipate that our findings should prove instrumental for potential applications of this technology to organic syntheses and drug discoveries in medicinal chemistry.

Transition metal-catalyzed cross-coupling reactions have been recognized as a powerful toolbox for the construction of C–C and C–X (X = heteroatoms) bonds, which play a vital role in the synthesis of pharmaceuticals, agrochemicals, and feedstock commodity chemicals[1–5]. For practical applications and easy availability in synthesis, oxygen-based electrophiles have been recognized as halide-free alternatives for coupling with organometallic reagents[6–10]. Among them, alkenyl esters represent an attractive candidate for non-aromatic functionality due to their versatile occurrence in biomass-derived chemicals and convenient preparation. However, most of the advances in cross-couplings were accomplished with activated esters, such as triflate, tosylate, sulfamate, carbamate, phosphate, and carbonate. Hence, numerous contributions have been made to further expand the library of electrophiles[11–16]. Alkenyl acetates[17–19], as a class of

(pseudo)halide electrophile bearing a small and nonhazardous leaving group, have received considerable recent attention[20]. While unactivated enol acetates in coupling reactions remain a significant challenge due to the selective cleavage issue between vinyl $Csp^2$–O and acetyl–O bonds (445 vs 335 kJ/mol)[21], using mild nucleophilic reagents as the coupling partners or/and devising highly active catalytic systems, which operate under kinetic control rather than a thermodynamic pathway, are the key for achieving cross-couplings via selective vinyl $Csp^2$–O bond cleavage (Fig. 1a).

Since Shi[22] and Garg[23] independently pioneered the feasibility of nickel-catalyzed Negishi- and Suzuki-coupling reactions by using alkenyl pivalates as the electrophiles, alkenyl acetates have been extensively exploited for transition metal-catalyzed C–C bond formation cross-coupling reactions with B, Zn, Mg-based organometallics

[1]Key Laboratory of Organic Synthesis of Jiangsu Province, Suzhou Key Laboratory of Pathogen Bioscience and Anti-infective Medicine, College of Chemistry, Chemical Engineering and Materials Science, Soochow University, Ren-Ai Road 199, Suzhou 215123, P. R. China. [2]Engineering Research Center of Organo-silicon Compounds & Materials, Ministry of Education, College of Chemistry and Molecular Sciences, Wuhan University, Wuhan, Hubei 430072, P. R. China. [3]School of Chemistry and Chemical Engineering, Henan Institute of Science and Technology, Xinxiang, Henan 453003, P. R. China. [4]State Key Laboratory of Elemento-Organic Chemistry, Nankai University, Tianjin 300071, P. R. China. ✉e-mail: jjackli@suda.edu.cn

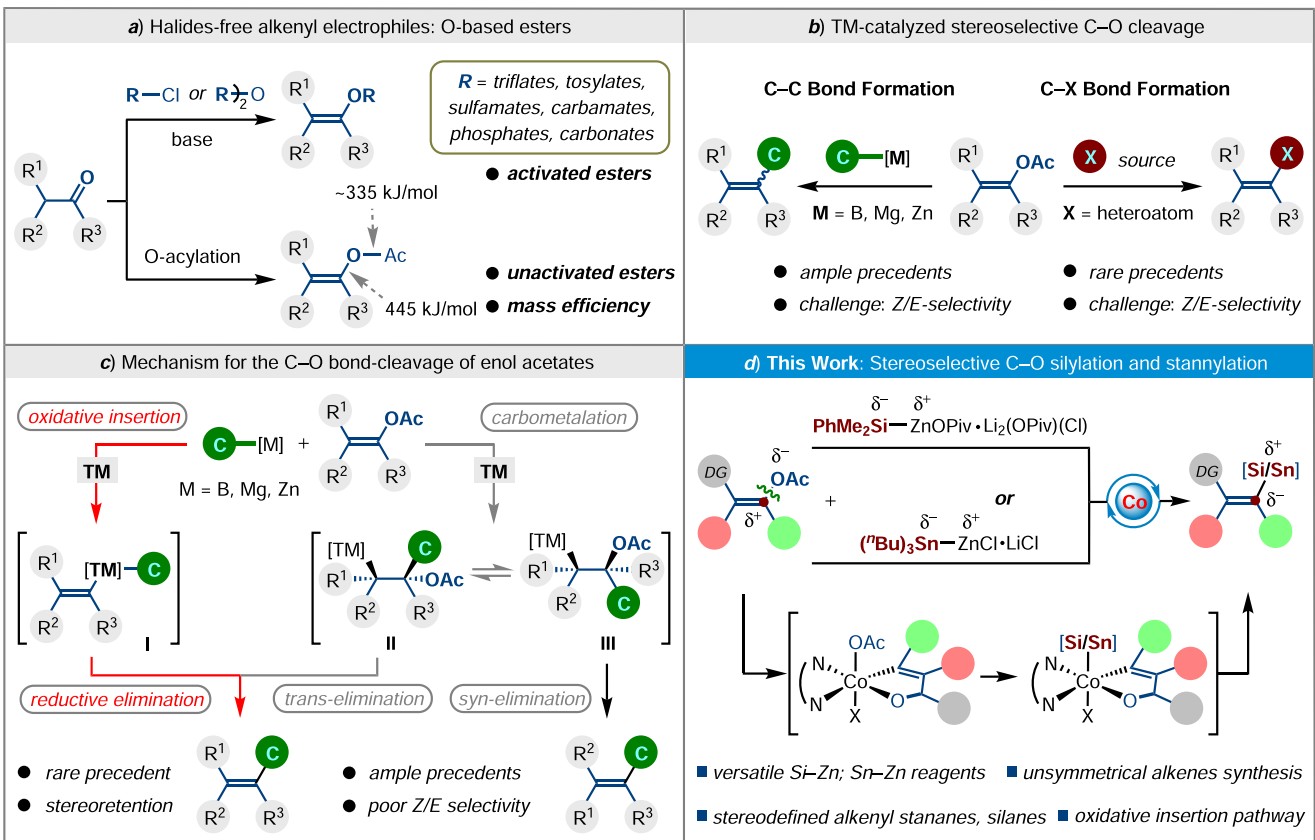

**Fig. 1 | Background of transition metal-catalyzed C−O bond activation. a** Oxygen-based electrophiles. **b** Transition-metal-catalyzed C−C and C−X bond formations through C−O bond cleavage of alkenyl acetates. **c** Mechanism for C−O bond cleavage of alkenyl acetates. **d** Cobalt-catalyzed stereoselective C−O silylation and stannylation.

using Ni[24], Fe[21], Co[25], Cr[26], Rh[27], and Pd catalysis[28–30]. Moreover, Ackermann[31] disclosed the chelation-assisted cobalt-catalyzed Csp²-H alkenylations of indoles with alkenyl acetates, which underwent a remarkably mild C−O bond cleavage as compared to the ruthenium catalysis[32]. Gosmini[33–35] and Shu[36] also reported the applications of vinyl acetates in Co- and Ni-catalyzed reductive arylation and alkylation. Despite these major advances, major challenges in the area of C−O functionalization with alkenyl acetates still need to be overcome: (i) in contrast to the common alkenyl C−C bond formation, rather rare examples of alkenyl C−heteroatom bond formation were developed;[37] (ii) most of the progresses have been made with cyclic, non- or mono-substituted alkenyl acetates as the coupling partners, rather poor Z/E-selectivity is observed when employing acyclic tri- or tetrasubstituted alkenyl acetates as the coupling partners. Hence, the methods aimed at C−heteroatom bond formation and stereo-controllable C−O cross-coupling are highly desirable (Fig. 1b).

Towards the mechanism of C−O bond activation, two plausible pathways have been given: initial transmetalation between organometallics and transition metal catalyst forms the C−[TM] species, which undergoes (i) oxidative insertion of C−O bond to the [TM]-center, along with following reductive elimination; or (ii) 1,2-carbometalation of enol carboxylates, in tandem with isomerization (**II** and **III**) and trans-, or syn-elimination. Therefore, avoiding the latter pathway, selective oxidative insertion to form a single diastereomer intermediate (**I**) is the key for achieving stereoselective C−O functionalization (Fig. 1c)[20]. We hypothesized that the challenging issue would be solved by introducing a removable directing group into the poly-substituted alkenyl acetates, thereby preferentially realizing chelation-assisted oxidative insertion of transition metal species to C−O bond, rather than 1,2-nucleophilic addition process. In this context, acyclic tri- and tetrasubstituted alkenyl acetates bearing a modulable carbonyl

group are attractive alternative candidates: OAc imparts better mass efficiency than other leaving groups, carbonyl groups are valuable functional groups and versatile precursors for alcohols, amines, and other important functionalities in organic synthesis. Organozinc reagents are important intermediates which display high reactivity and excellent chemoselectivity in organic synthesis[38–44]. Hence, we developed the bench-stable and easy-to-handle silyl and stannylzinc reagents[45,46], and applied to 3d cobalt-catalyzed cross-couplings[47–53] with polyfunctionalized alkenyl acetates to build synthetically useful C−Si[54–56] and C−Sn[57–60] bonds (Fig. 1d). In this work, Me₂PhSi–ZnOPiv•Li₂(Cl)(OPiv) and (ⁿBu)₃Sn–ZnCl•LiCl show superior reactivity over the other commonly available Si- and Sn-based organometallic reagents. Moreover, notable features of our strategy are excellent functional group tolerance, and complete control of regio- and stereoselectivity, thus providing expedient access to the stereodefined alkenyl silanes and stannanes, which can be further utilized for the synthesis of stereochemically defined acyclic tri- and all-carbon tetrasubstituted alkenes.

## Results

### Optimization studies

We initiated our studies by preparing different Si–Met (**1**) and Sn–Met (**2**) reagents (Met = Li, Mg, Zn, and Al) through transmetalation of corresponding Si–Li and Sn–Li reagents with MgBr₂, ZnX₂, and Zn(OPiv)₂, or AlCl₃. Thereafter, we tested their reactivity for the cobalt-catalyzed silylation and stannylation of alkenyl acetates via C−O bond activation (Fig. 2). To our delight, CoBr₂ in the presence of 2,2′-bipyridyl enables silylation of **3a** using Me₂PhSi–ZnOPiv•Li₂(Cl)(OPiv) (**1a**) as the nucleophile [PhMe₂Si–ZnOPiv•2Li(OPiv)(Cl) was abbreviated as Me₂PhSi–ZnOPiv for the sake of clarity], which displayed unique reactivity in carbonsilylation of alkenes by nickel catalysis[45], the

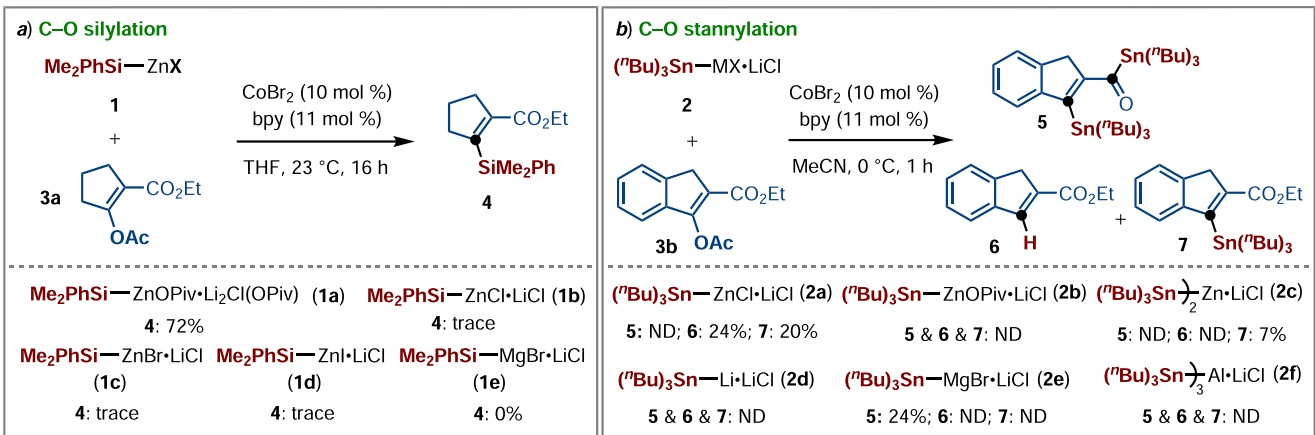

**Fig. 2 | Cobalt-catalyzed C–O bond cleavage with different Si–Zn (1) and Sn–Zn (2). a** Cobalt-catalyzed C–O silylation of **3a**. **b** Cobalt-catalyzed C–O stannylation of **3b**. Reaction conditions: **3a** or **3b** (0.25 mmol, 1.0 equiv), **1** or **2** (0.325 mmol, 1.3 equiv), CoBr$_2$ (10 mol%), 2,2′-bipyridine (bpy; 11 mol%), THF or MeCN (1.5 mL), 23 or 0 °C, isolated yields.

desired product **4** was obtained in 72% yield. Interestingly, the typical halides-supported Si–Zn reagents **1b–1d**, as well as Si–Mg reagent **1e** failed to undergo the C–O silylation process under the identical cobalt catalysis. These findings are consistent with the previous nickel-catalyzed carbosilylation of alkenes, which demonstrated the crucial importance of OPiv-coordination for improving the reactivity of Si–Zn reagents. Furthermore. cross-coupling product was not observed in the treatment of Sn–Li (**2d**), Sn–Zn (**2b**), or Sn–Al (**2 f**) with indenyl acetate (**3b**) in the presence of CoBr$_2$/2,2′-bipyridine (10 mol%) in MeCN at 0 °C. Whereas a two-step sequence including cobalt-catalyzed stannylation and nucleophilic addition/elimination occurred when employing Sn–Mg (**2e**) as the nucleophile, affording the bis-stannylated product **5** in 24% yield. The envisioned alkenyl stannane **7** was obtained using Sn–Zn **2a** or **2c** as the nucleophiles, and thus ($^n$Bu$_3$) Sn–ZnCl•LiCl yielding the desired product **7** in 20% yield, along with 24% of the destannylative product **6**.

These results encouraged us to further optimize the reaction conditions for cobalt-catalyzed C–Si and C–Sn bonds formation using Me$_2$PhSi–ZnOPiv•Li$_2$(Cl)(OPiv) (**1a**) and ($^n$Bu)$_3$Sn–ZnCl•LiCl (**2a**) as the nucleophilic partners (Table 1). Firstly, optimization studies identified THF as the ideal reaction medium and bipyridine (**L1**) or 5,5′-dimethylbipyridine (**L4**) as the ligands of choice for the cobalt-catalyzed C–O silylation, leading to the desired alkenyl silane **4** in 81% yield (Table 1a, entries 1–8). Lowering the temperature or employing other cobalt complexes as the catalyst gave moderate yields (entries 9–12). However, examination with a metal source such as nickel, iron, or chromium salts provided only a negligible conversion, as was also observed in the absence of a catalyst (entries 13–14). After extensive optimization of ligands, such as bis(pyridine)s (**L1–L7**), tridentate 2,6-bis(N-pyrazolyl)pyridine (**L8**) phosphines (**L9–L11**), and bipyridine (**L1**) gave the optimal conversion of **7** (Table 1b, entries 1–11). Meanwhile, MeCN was identified as the ideal medium of choice (entry 12). Switching from CoBr$_2$ to NiCl$_2$, FeCl$_2$, or CrCl$_2$ failed to improve the transformation (entry 13). A significant breakthrough was made when performed the reaction at −20 °C, affording **7** in 53% yield, while no reaction was observed at −30 °C (entry 14). Notably, shortening the reaction time to 15 min led to a further yield increase to 66% yield (entries 15–16), which highlighted the robustness of the cobalt catalyst and unique reactivity of ($^n$Bu)$_3$Sn–ZnCl•LiCl reagent (**2a**).

## Substrate scope studies
With the optimized cobalt catalyst in hand, the generality of our protocol was subsequently explored in a range of C–O bond-silylation reactions with various polyfunctionalized alkenyl acetates **5** and solid

silylzinc pivalate PhMe$_2$Si–ZnOPiv (**1a**) (Fig. 3). We were pleased to find that cyclic and various all-carbon tetrasubstituted acyclic alkenyl acetates smoothly proceeded cobalt-catalyzed C–O bond-silylation with **1a** under the otherwise identical reaction conditions. The solid PhMe$_2$Si–ZnOPiv (**1a**) exhibited good reactivity under remarkably mild reaction conditions at 23 °C, thereby delivering the corresponding tetrasubstituted alkenyl silanes **9–30** in 50–86% yields. Importantly, the cobalt catalysis displayed excellent functional group compatibility, such as fluoro, chloro, bromo, ester, nitrile, acetate, piperidine, and alkenyl substituents, were well tolerated under the mild reaction conditions. Notably, the catalytic system showed excellent chemoselectivity to realize C–O bond activation, rather than C–Hal (Cl and Br) cross-couplings. Moreover, the C–O bond cleavage of all-carbon tetrasubstituted acyclic alkenyl acetates **5** smoothly underwent a silylation process in a highly stereoretentive fashion, the corresponding (Z)-alkenyl silanes **9–31** were obtained with excellent stereoselectivity. A similar result was also observed when employing (Z)-alkenyl acetate bearing a lactone fragment as the electrophile, furnishing the (Z)-stereoisomer **32** as the sole product in 81% yield, which could be further reduced to the ring-opening diol **33** in a stereoretentive fashion. In sharp contrast, the lactam substrate failed to deliver the silylated product. The trisubstituted acyclic acetates failed to generate the desired product because a significant amount of the corresponding hydrodesilylated product **34** was formed, which is likely owing to the poor stability of the alkenyl silane under the reaction conditions.

On the basis of the remarkably high stereoselectivity and efficacy of the C–O bond-silylations, we subsequently succeeded in examing analogous stannylations of various alkenyl acetates with ($^n$Bu$_3$)Sn–ZnCl•LiCl (**2a**). Indeed, a variety of cyclic enol acetates, including indenyl acetate, naphthalenyl acetate, chromenyl acetate, as well as cyclopentenyl and cycloheptenyl acetates, were smoothly stannylated with **2a**, delivering the alkenyl stannanes **7**, **35–39** in 47–85% yields. Moreover, the desired C–O bond stannylation can be successfully scaled up to the gram scale with a slightly reduced yield (**38**) (Fig. 4a).

Furthermore, we turned our attention to examining the stereoselective transformation of our cobalt catalysis. It is worth noting that the (Z)-alkenyl acetates underwent cobalt-catalyzed stannylation in a stereoretentive fashion, generating the (Z)-stereoisomer as the sole product. As expected, various trisubstituted acrylates bearing electron-rich as well as electron-deficient substituents on the aryl side were identified as viable substrates for direct stannylations under remarkably mild reaction conditions, furnishing the (Z)-diastereomers **40–51** in 49–78% yields through excellent chemo- and stereoselective C–O bond activation. Moreover, a variety of valuable functional groups such as trifluoromethyl, ester, nitrile, and terminal alkene were well

**Table 1 | Optimization of the reaction conditions**

Co-catalyzed C–O silylation:
PhMe₂Si—ZnOPiv•Li₂(OPiv)(Cl) (**1a**, 2.0 equiv), CoBr₂ (10 mol %), **L1** (11 mol %), THF, 23 °C, 2 h. **3a** → **4** (SiMe₂Ph)

Co-catalyzed C–O stannylation:
(ⁿBu)₃Sn—ZnCl•LiCl (**2a**, 2.0 equiv), CoBr₂ (10 mol %), **L1** (11 mol %), MeCN, 0 °C, 1 h. **3b** → **7** (Sn(ⁿBu)₃)

Ligands Screening: L1 (R = H), L2 (R = tBu), L3 (R = OMe); L4; L5 (R = H), L6 (R = Ph); L7; L8; L9; L10; L11

**Co-catalyzed C–O silylation (3a to 4)[a]**

| Entry | Modifications | Yield (%)[b] |
|---|---|---|
| 1 | none | 81 |
| 2 | **L2** instead of **L1** | 40 |
| 3 | **L3** instead of **L1** | trace |
| 4 | **L4** instead of **L1** | 80 (81)[c] |
| 5 | **L5** instead of **L1** | trace |
| 6 | **L7** instead of **L1** | trace |
| 7 | **L11** instead of **L1** | trace |
| 8 | PhMe or NMP or MeCN | 0–20 |
| 9 | 0 °C | 57 |
| 10 | CoI₂ instead of CoBr₂ | 55 |
| 11 | CoCl₂ instead of CoBr₂ | 62 |
| 12 | Co(acac)₂ instead of CoBr₂ | 33 |
| 13 | NiCl₂ or FeCl₂ or CrCl₂ | 0–8 |
| 14 | w/o [Co] | 0 |

**Co-catalyzed C–O stannylation (3b to 7)[a]**

| Entry | Modifications | Yield (%)[b] |
|---|---|---|
| 1 | none | 20 |
| 2 | **L2** instead of **L1** | 12 |
| 3 | **L3** instead of **L1** | trace |
| 4 | **L4** instead of **L1** | 18 |
| 5 | **L5** instead of **L1** | 18 |
| 6 | **L6** instead of **L1** | 15 |
| 7 | **L7** instead of **L1** | 0 |
| 8 | **L8** instead of **L1** | trace |
| 9 | **L9** instead of **L1** | 14 |
| 10 | **L10** instead of **L1** | 18 |
| 11 | **L11** instead of **L1** | 20 |
| 12 | THF or DMF or dioxane | 0 |
| 13 | NiCl₂ or FeCl₂ or CrCl₂ | 0–8 |
| 14 | −20 °C (or −30 °C) | 53 (0) |
| 15 | −20 °C, 25 min | 54 |
| 16 | −20 °C, 15 min | 66[d] |

*NMP* N-methyl pyrrolidone, *DMF* N,N-dimethylformamide.
[a] Reaction conditions: **3a** or **3b** (0.25 mmol, 1.0 equiv), **1a** (0.50 mmol, 2.0 equiv), or **2a** (0.325 mmol, 1.3 equiv), CoBr₂ (10 mol %), **L1** (11 mol %), THF or MeCN (1.5 mL), 0 or 23 °C.
[b] Isolated yields.
[c] CoBr₂(5,5'-dmbpy) (10 mol%) as the catalyst.
[d] Along with 21% of the product **6**.

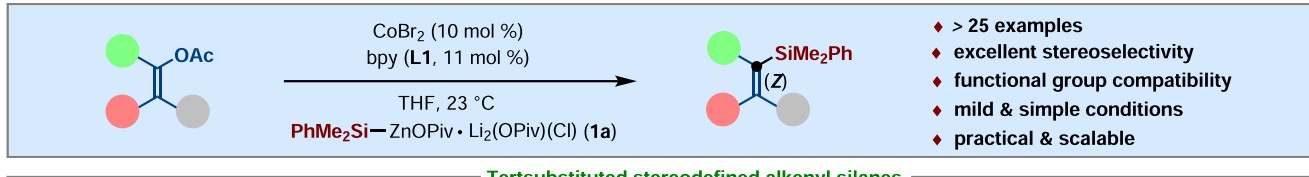

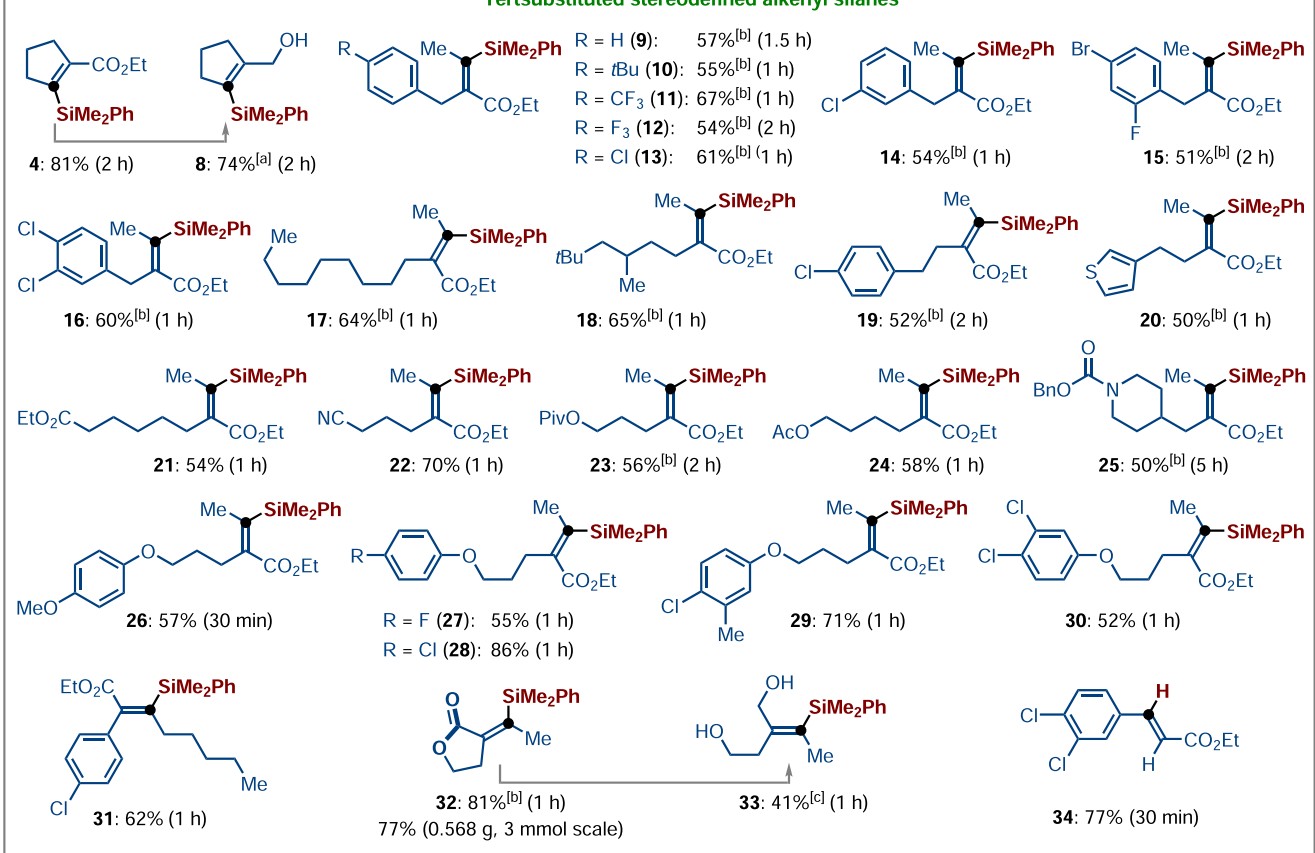

**Fig. 3 | Substrate scope for stereoretentive C–O bond silylation.** Reaction conditions for silylation: alkenyl acetates (0.25 mmol, 1.0 equiv), Me₂PhSi–ZnOPiv **1a** (0.50 mmol, 2.0 equiv), CoBr₂ (10 mol %), bpy (**L1**, 11 mol %), THF (1.5 mL), 23 °C. [a] DIBAH (2.0 equiv), THF, −78 °C. [b] CoBr₂(5,5′-dmbpy) (10 mol%) as the catalyst. [c] LiAlH₄ (1.5 equiv), Et₂O, 0 °C. DIBAH diisobutylaluminum hydride.

tolerated under otherwise identical reaction conditions. A similar result was also observed when employing (Z)-acetoxyheptenyl carboxylate as the electrophilic partner, giving the desired product **52** in 76% yield (Fig. 4b).

Toward the stereoselective preparation of more challenging unsymmetrical all-carbon tetrasubstituted alkenyl stannanes, a series of stereodefined all-carbon tetrasubstituted alkenyl acetates were investigated. To our delight, a range of all-carbon tetrasubstituted alkenyl acetates bearing different functional groups smoothly underwent the cobalt-catalyzed C–O bond stannylation to afford the desired products **53–68** in 47–84% yields with complete control of stereoselectivity. It is worth noting that tertiary amide, lactam, as well as lactone were found to be effective chelation groups for the envisioned stereoselective C–O bond-stannylation reaction, thereby providing (Z)-selective **69–72** as the sole products (Fig. 4b).

In order to further illustrate the potential applications of our protocol in the late-stage modifications of natural products and pharmaceutically active molecules. The alkenyl acetates derivatives of cholesterol, lithocholic acid, and citronellol were proven to be viable substrates for cobalt-catalyzed C–O silylation with PhMe₂Si–ZnOPiv (**1a**), delivering the corresponding alkenyl silanes **73–76** in moderate yields with high (Z)-selectivity. Moreover, the enol acetates derivatives of citronellol, ibuprofen, probenecid, celecoxib, and indomethacin

were well stannylated with (ⁿBu)₃Sn–ZnCl•LiCl, leading to the (Z)-diastereomers **77–81** in 47–71% yields, thus displaying the potential utility of this technology to drug discovery in medicinal chemistry (Fig. 5).

**Mechanistic studies**
Inspired by the high reactivity of the PhMe₂Si–ZnOPiv (**1a**) and (ⁿBu)₃Sn–ZnCl•LiCl (**2a**) for cobalt-catalyzed stereoselective C–O bond silyl- and stannylation of alkenyl acetates, we sought to unravel the reaction mechanism. To this end, a series of control experiments with radical scavengers **82** were performed, and both silylated and stannylated products were isolated with slightly reduced yields (Fig. 6a). We next set the parallel reactions with stereodefined (Z)- and (E)-alkenyl acetates, substrate (Z)-**83** afforded the stereoretentive product **84** in 86% yield, whereas the resulting products were detected by GC analysis with an E/Z ratio of 5.4:1 when employing (E)-**83** as the electrophile. Moreover, a (E)-alkenyl acetate (**85**) bearing a conjugated amide only delivered the desired product **86** in a negligible conversion (Fig. 6b). These findings suggested that the chelation-assisted oxidative insertion of cobalt into the C–O bond of (Z)-alkenyl acetates is preferred to (E)-alkenyl acetates. Intermolecular competition experiments with (Z)-and E-alkenyl acetates revealed (Z)-**83** to be inherently more reactive (Fig. 6c). Switching from (Z)-**87** to the more hindered substrate of (Z)-**88** led to a significantly decreased yield of the

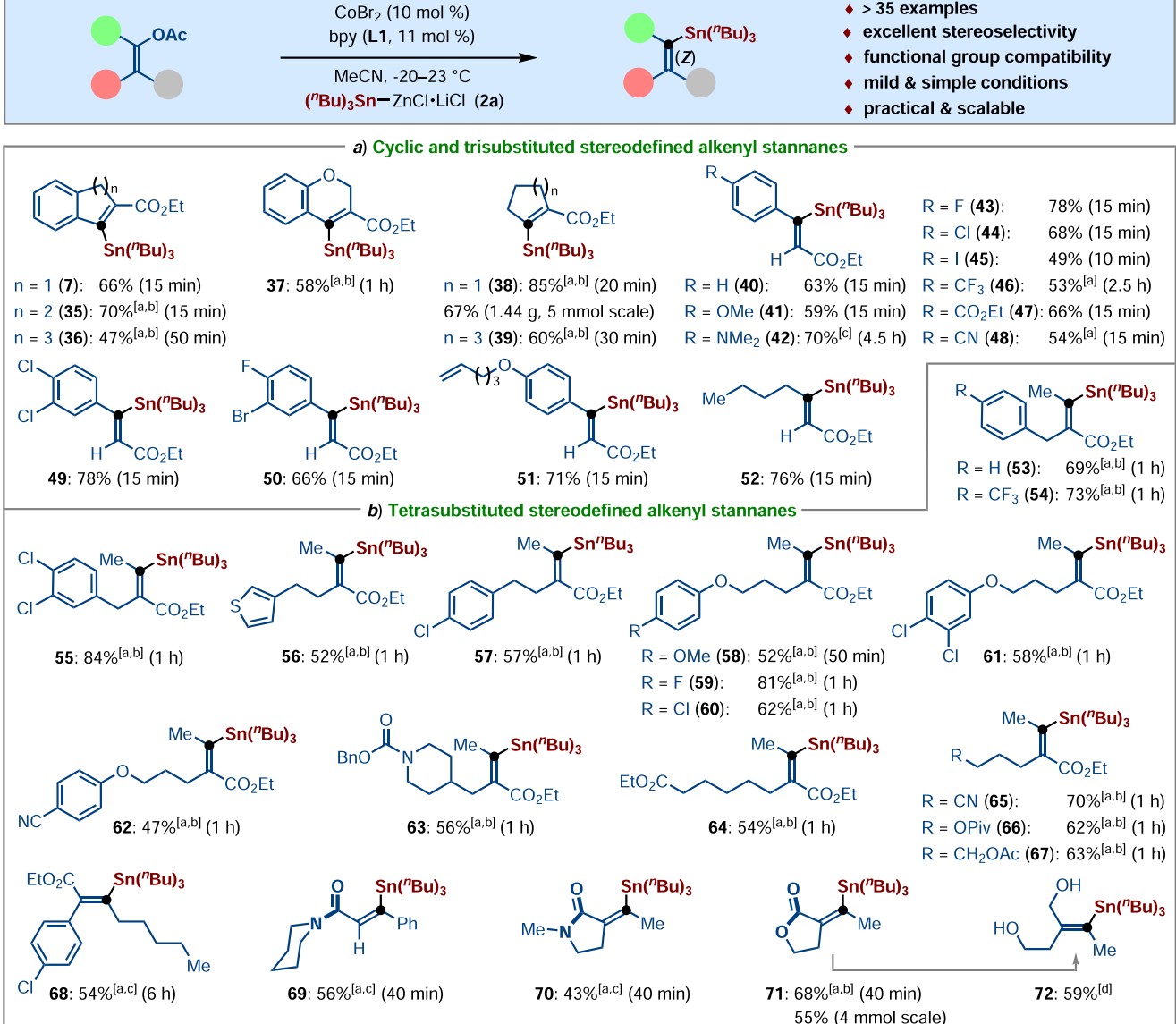

**Fig. 4 | Substrate scope for stereoretentive C–O bond stannylation. a** Scope of C–O stannylation with cyclic and trisubstituted alkenyl acetates. **b** Scope of C–O stannylation with tetrasubstituted alkenyl acetates. Reaction conditions for stannylation: alkenyl acetates (0.25 mmol, 1.0 equiv), **2a** (0.325 mmol, 1.3 equiv), CoBr₂ (10 mol%), bpy (**L1**, 11 mol%), MeCN (1.5 mL), −20 °C. [a] CoBr₂ and **L1** (15 mol%). [b] 0 °C. [c] 23 °C. [d] LiAlH₄ (1.5 equiv), Et₂O, 0 °C.

corresponding product **89** and **90** (0 and 31%) (Fig. 6d). In addition, the substrate bearing a conjugated *N,N*-dimethyl amide (Z)-**91** only gave the desired stannane **93** in 43% yield with complete control of geometry. Moreover, an enol acetate **94** bearing a conjugated nitrile was also examined under standard conditions. However, only a trace amount of silylated product **95** and a low yield of stannylated product **96** were formed. As expected, no desired C–O bond functionalization was observed when using the substrate without a conjugated ester group as the electrophilic coupling partner (see Supplementary Fig. 1e). These findings suggested that the ester group is crucial to the stereoselective C–O bond activation due to its chelation-assistance and conjugation effect to active the α,β-unsaturated alkenyl acetate system (Fig. 6e).

Based on our mechanistic studies and previous insights[26,29], a plausible mechanism for the cobalt-catalyzed stannylation of alkenyl acetates is proposed, as shown in Fig. 7. The Co(II) precatalyst is initially reduced to form an active $L_nCo^I$-species by the [Si/Sn]–ZnX (**1a** or **2a**). Alkenyl acetate **3** then coordinates to the in situ formed $L_nCo^I$-species (**A**) with the assistance of ester chelation. The chelation effect

can facilitate the oxidative addition of $Co^IL_n$ into C–O bond, thus forming Co(alkenyl)(OAc) complex **B** as a single diastereomer. Subsequent coordination and transmetalation with [Si/Sn]–ZnX (**1a** or **2a**) affords the intermediate **D**. Final reductive elimination of **D** releases the desired alkenyl silanes and stannanes in a stereoretentive fashion and regenerates the catalytically active cobalt species **A** to close the catalytic cycle.

Finally, the synthetic utility of the resulting alkenyl silanes and stannanes was further described in the palladium-catalyzed Hiyama and Stille coupling reactions (Fig. 8). Remarkably, aryl halides bearing chloro, and sensitive ketone, aldehyde substituents were well coupled with various alkenyl silanes and stannanes under mild reaction conditions, delivering the arylated products **97**–**100** in moderate to excellent yields with complete control of stereoselectivity, the conformation of **100** was confirmed by X-ray crystal analysis. 1,3-Dienes are key important structural motifs of natural products with activities of relevance to biology chemistry[61,62]. Such building blocks could be expediently constructed by palladium-catalyzed stereoselective alkenyl–alkenyl coupling reactions; the desired 1,3-dienes **101**–**105** were

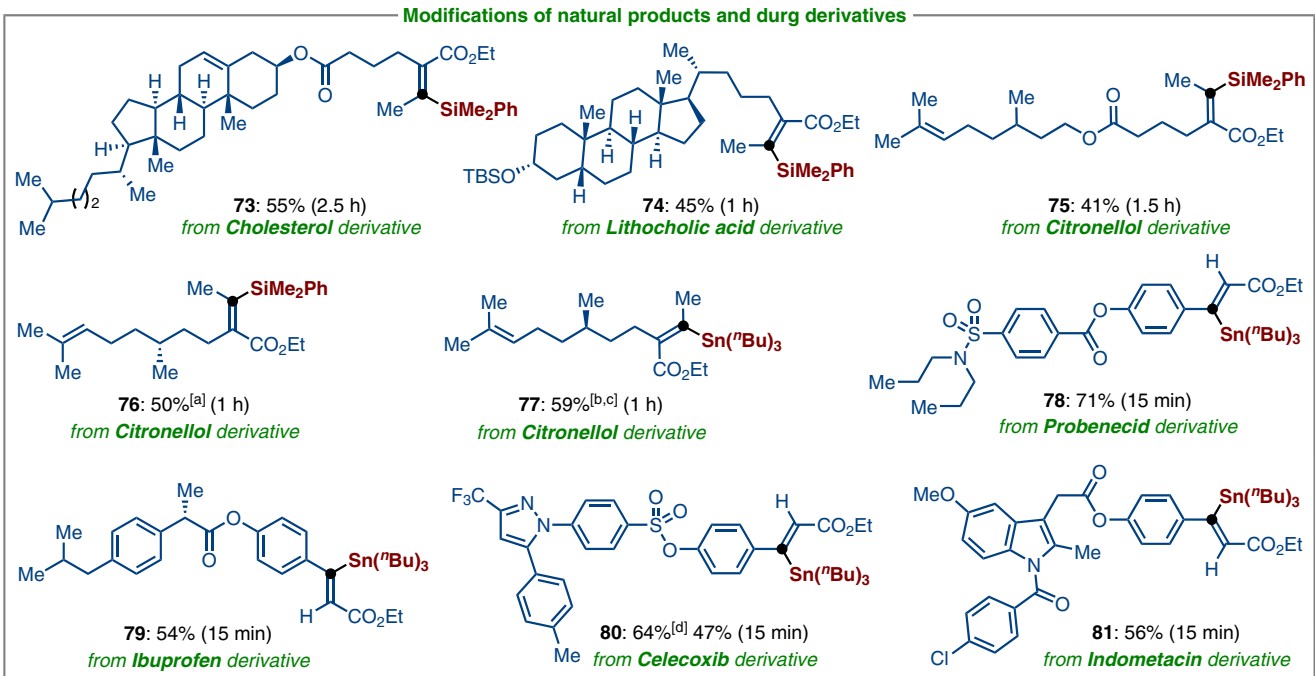

**Fig. 5 | Late-stage functionalization of drugs and natural products by stereo-retentive C−O bond cleavage.** Reaction conditions for silylation: alkenyl acetates (0.25 mmol, 1.0 equiv), Me₂PhSi–ZnOPiv **1a** (0.50 mmol, 2.0 equiv), CoBr₂ (10 mol%), bpy (**L1**, 11 mol%), THF (1.5 mL), 23 °C. Reaction conditions for stannylation: alkenyl acetates (0.25 mmol, 1.0 equiv), **2a** (0.325 mmol, 1.3 equiv), CoBr₂ (10 mol%), bpy (**L1**, 11 mol%), MeCN (1.5 mL), −20 °C. [a] CoBr₂(5,5′-dmbpy) (10 mol%) as the catalyst. [b] CoBr₂ and **L1** (15 mol%). [c] 0 °C. [d] 23 °C.

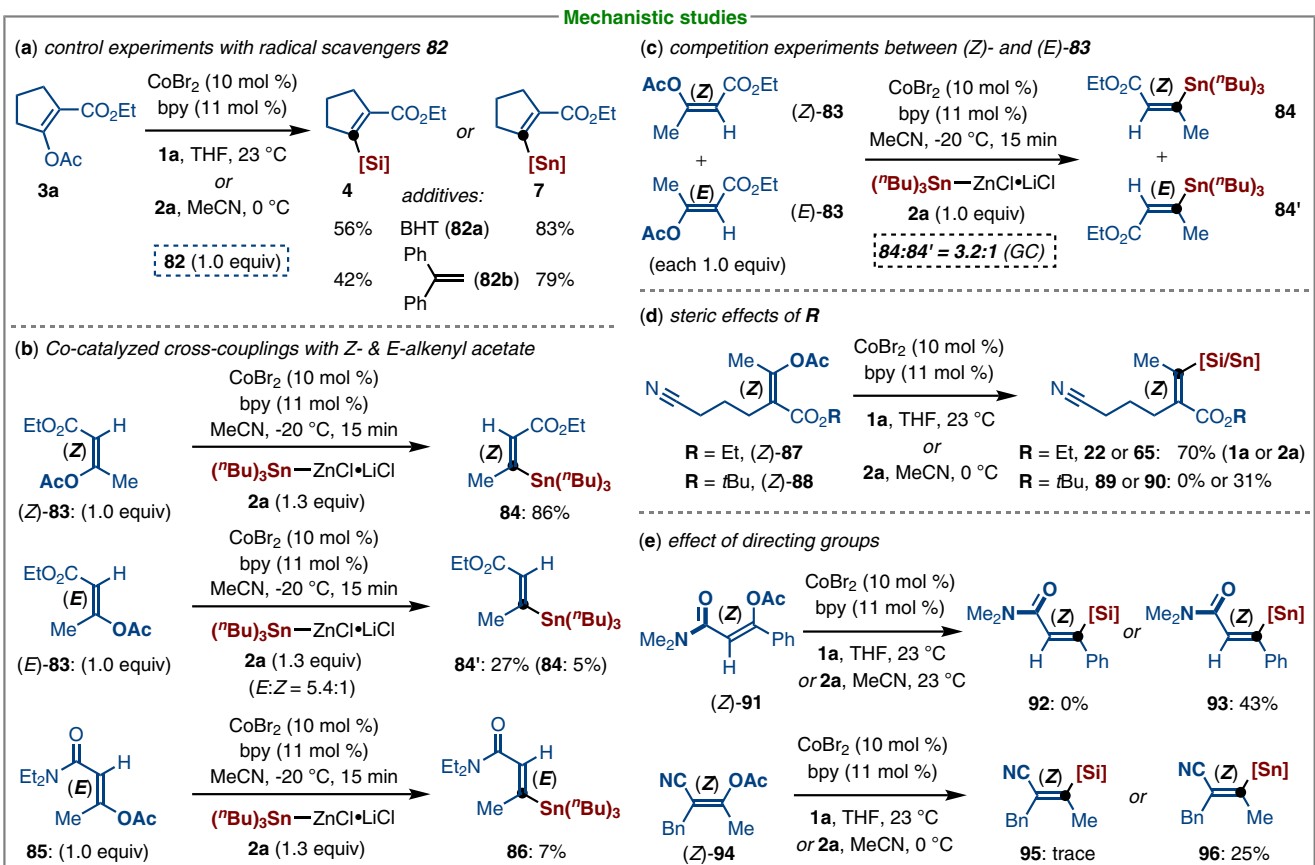

**Fig. 6 | Mechanistic studies. a** Control experiments with radical scavengers. **b** Investigation of cobalt-catalyzed stereoselective C−O bond cleavage. **c** Competition experiments of stereodefined alkenyl acetates. **d** Steric effects of different esters. **e** Chelation-assistance of directing groups.

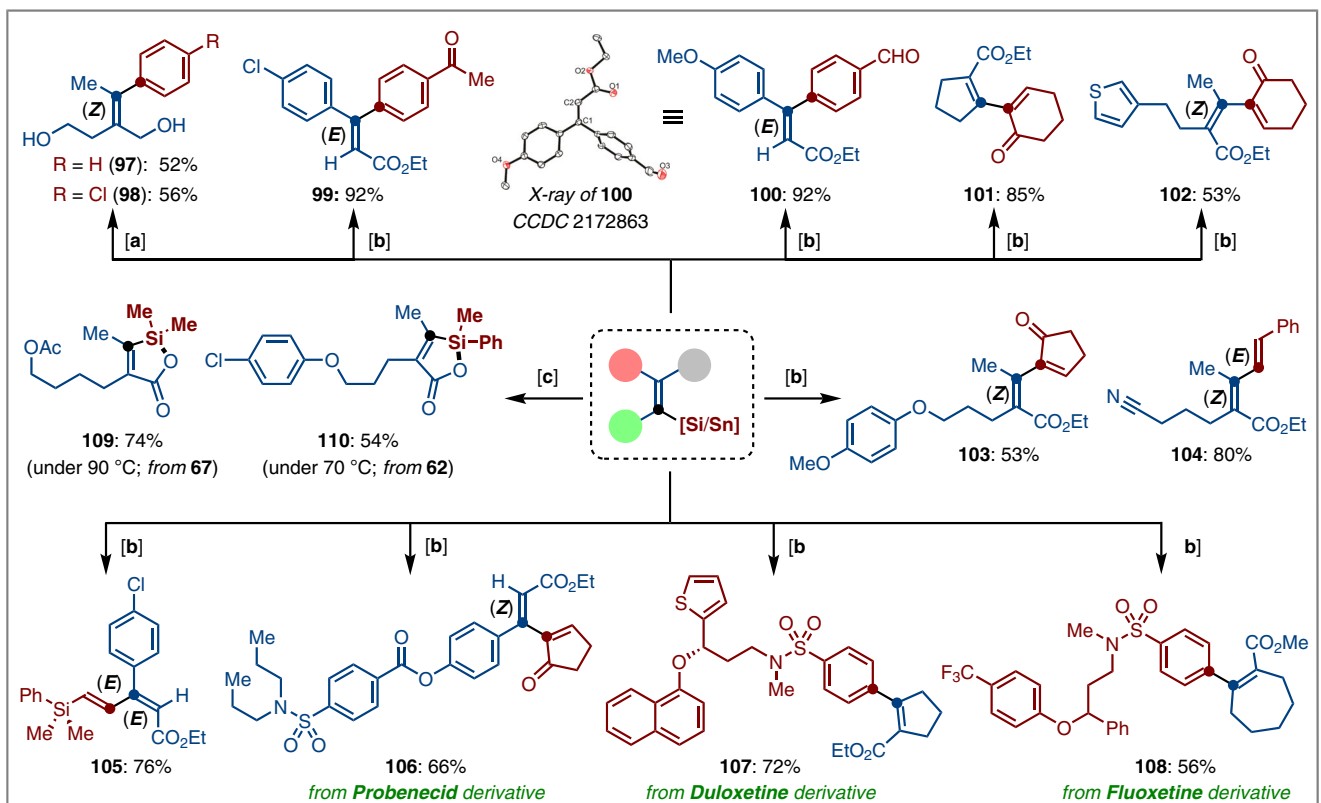

**Fig. 7 | Catalytic cycle.** A plausible mechanism for the chelation-assisted C–O silylation and stannylation.

**Fig. 8 | Facile late-stage functionalizations.** [a] Reaction conditions: electrophiles (**Ar**–I; 2.0 equiv), alkenyl silanes (1.0 equiv), Pd$_2$(dba)$_3$ (2.5 mol%), NaOH (2.0 equiv), THF, 60 °C, 16 h. [b] Reaction conditions: electrophiles (**E**-I/Br; 1.2 equiv), alkenyl stannanes (1.0 equiv), Pd(PPh$_3$)$_4$ (5 mol%), CuI (50 mol%), DMF, 23 °C, 16 h. [c] I$_2$ (1.5 equiv), CCl$_4$, 90 or 70 °C.

afforded in a stereoselective fashion. Beyond that, the potential applications of our protocol to the late-stage modifications of drug-like molecules, such as probenecid, duloxetine, and fluoxetine derivatives, were further demonstrated by a facile Stille-alkenylation under

room temperature to afford the desired products **106–108** in 56–72% yields, thus proving instrumental technology for the synthesis of highly complex molecules in drug discovery. Moreover, these alkenyl silanes can be further converted to the corresponding silalactones

**109**–**110** through selective Si–C bond electrophilic cleavage. Interestingly, the selectivity was largely dominated by the reaction temperature.

## Discussion

In conclusion, we have demonstrated the utility of Si–Zn and Sn–Zn reagents for the efficient cobalt-catalyzed cross-coupling of alkenyl acetates through chelation-assisted C–O bond cleavage under mild reaction conditions, thus providing a practical and efficient strategy for the preparation of various tri- and tetrasubstituted alkenyl silanes and stannanes in a stereoretentive fashion. Indeed, the solid silylzinc pivalate and stannylzinc chloride showed excellent reactivity and chemoselectivity toward diverse alkenyl acetates, as well as an ample substrate scope and high functional group compatibility. Preliminary mechanism studies suggested that the stereoselectivity presumably is caused by the chelation-assisted oxidative addition of cobalt to the C–O bond of alkenyl acetates. Finally, we have also illustrated the synthetical versatility of the polyfunctionalized alkenyl silanes and stannanes by the facile transformations of Hiyama and Stille reactions with complete control of stereoselectivity, such as arylation and alkenylation. Further development of the related applications with Si–Zn and Sn–Zn reagents are currently ongoing in our laboratory and will be reported in due course.

## Methods

### Cobalt-catalyzed stereoselective C–O silylation and stannylation of alkenyl acetates

To an oven-dried 25 mL screw-cap tube equipped with a stir bar was vacuumed and backfilled with $N_2$ for three times. A suspension of $CoBr_2$ (10 mol%), 2,2′-bipyridyl (11 mol%) in THF or MeCN (0.5 mL) was charged and stirred at 0 °C for 5 min, then the alkenyl acetate (0.25 mmol, 1.0 equiv), and $PhMe_2Si–ZnOPiv·Li_2(Cl)(OPiv)$ (1a, 0.5 mmol, 2.0 equiv, 1.0 mL THF) or $(^nBu_3)Sn–ZnCl·LiCl$ (2a, 0.325 mmol, 1.3 equiv, 1.0 mL MeCN) were sequentially added to the catalyst suspension. The reaction mixture was stirred at the identified temperature (silylation: 23 °C; stannylation: −20, or 0, or 23 °C) under an atmosphere of $N_2$. At ambient temperature, the solvent was evaporated under reduced pressure and the remaining residue was purified by column chromatography on silica gel to yield the desired products.

## Data availability

The experimental procedures and compound characterization, and related data generated in this study are provided in the Supplementary Information/Source Data file. The X-ray crystallographic data of 100 used in this study are available in the joint Cambridge Crystallographic Data Centre and Fachinformationszentrum Karlsruhe Access Structures service www.ccdc.cam.ac.uk/structures.

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

## Acknowledgements

We thank the Natural Science Foundation of Jiangsu Province (BK20221355), the start-up grant (NH10900422) of Soochow University, the sponsored program by the Jiangsu Province for the Cultivation of Innovation and the Fundamental Research Funds for the Central Universities (WHU 2042022kf1038 and 2042021KF1020) for financial support.

## Author contributions

Jie. L. conceived and directed the project; Jie. L. and X.Q. wrote the paper; Y.H. and B.H. designed and performed the experiments; X.Q. and J.P. participated in the design and discussion of the mechanism; J.W., J.J., J. Lin, and X.L. prepared the starting materials; all the authors were involved in the interpretation of the results presented in the manuscript.

## Competing interests

The authors declare no competing interests.
