## [Peer Review File · Nature Communications]

Stereoselective C–O Silylation and Stannylation of Alkenyl AcetatesREVIEWER COMMENTS

Reviewer #1 (Remarks to the Author):

The submitted manuscript from Qi, Xi, and co-workers focuses on a cobalt-catalyzed method for the substitution of vinyl acetate groups with silyl and stannyl nucleophiles. This results in the formation of new vinyl silanes and stannanes in a stereoselective manner, where the alkene stereochemistry is retained between starting material and product. The catalyst system - CoBr₂ and bipyridine - is simple, inexpensive, and readily available. The main group nucleophiles are somewhat more laborious to prepare, requiring formation of the lithium reagents from the silyl/stannyl chlorides, followed by transmetalation with ZnX₂. The reaction scope presented is very impressive, and products yields are synthetically useful. The products obtained (i.e. vinyl silanes/stannanes) are not themselves directly useful, but rather have potential as versatile intermediates for further functionalization/elaboration en route to a more complex target. The authors demonstrate this point with some examples of further reactivity in Figure 8, mostly involving Stille coupling of the vinyl stannane derivatives.

Overall, this is a highly significant advance in synthetic methodology. Achieving selective activation/substitution of vinyl acetate derivatives is very challenging, due both to the strength of the C-O bond, and the myriad side reactions these substrate are prone to, such as competitive nucleophilic attack at the carbonyl carbon of the acetate group. Moreover, acyclic vinyl acetates provide an additional challenge in controlling the stereochemical outcome. While vinyl silane/stannane synthesis is perhaps less valuable than a direct substitution with carbon-based nucleophiles, the possibility of further reactivity means this is a potentially useful approach. The demonstrated reaction scope and simplicity of the catalyst system are particular strengths of this work.

This being said, there are several aspects of this work that should be addressed before publication is considered:

-The new compounds all appear to be characterized by 1D NMR and MS methods (and in one specific case, XRD), though other methods determining composition and/or purity are not used (e.g. IR spectroscopy, melting points for solids, HPLC or GC traces for purity). While these alternative techniques are probably not necessary, there is the question of how the authors confirmed the product alkene stereochemistry for the acyclic products. There does not appear to be any mention/discussion of how this was accomplished (e.g. by NOE NMR techniques), nor any primary data in the SI addressing this.

-For the "mechanistic studies" in Figure 5, I'm not sure of the value of the "radical clock" experiment in 5a. Since the silylation and stannylation of this (E)-vinyl acetate do not actually seem to occur, then no conclusions can be made about 1-electron versus 2-electron pathways. Can the authors devise a more conclusive experiment to probe for radical intermediates?

-To further probe the hypothesized chelation assisted oxidative addition, can the authors access a substrate where the ester group has been replaced by a nitrile? This would give a similarly resonance-activated vinyl acetate, but lack the key carbonyl oxygen for coordination to the Co center. The differential reactivity between (Z) and (E) substrates observed (e.g. in Fig 5b and 5c) could be due to electronic differences rather than chelation assistance. Can the authors also computationally evaluate the predicted/calculated electrophilicity of (Z)-85 versus (E)-85? Perhaps the (Z) isomer is inherently more electrophilic, and that chelation assistance is not required to explain the reactivity.

-For the calculated reaction pathway, did the authors similarly evaluate other potential pathways computationally? For example, an oxidative addition based mechanism where there is no chelation assistance, or the alkene insertion / beta-acetoxy elimination pathway? Also, please note that I am by no means a computational chemistry expert, and so I cannot comment on the computational methods used (another reviewer who is a computational expert should be consulted for this).

-One additional paper that should be cited in the context of vinyl acetate substitution using main-group reagents is: *New J. Chem.*, 2021, 45, 20095-20098 (borylation of vinyl acetates and pivalates using Pd catalysis, albeit with much more limited scope).

-Finally, the manuscript text requires thorough editing/revision for grammar and diction. I will not itemize the needed changes here, as there are many (I count at least 10 in the first paragraph alone). As-written, the text errors are a significant distraction from the work itself. Given the high quality of the synthetic method, the authors no doubt would want a high quality text presentation as well.

In summary, this work could certainly become suitable for publication in *Nature Communications*, provided the details on stereochemical confirmation are included, some additional clarity on the mechanism is provided, and the text itself is improved.

Reviewer #2 (Remarks to the Author):

Qi, Li and co-workers describe cobalt-catalyzed cross-couplings of alkenyl acetates with silyl- and stannylzinc reagents to provide access to synthetically valuable polysubstituted alkenyl silanes and stannanes, respectively. This work builds on an earlier report of one of the authors in which the cross-coupling of alkenyl acetates with aryl or alkenyl zinc pivalates to form C(sp²)-C(sp²) bonds was realized under identical reaction conditions (see Ref. 25). Key to success for the unprecedented silylation and stannylation reactions is the use of the solid silylzinc pivaloate Me₂PhSiZnOPiv•Li₂(Cl)(OPiv) and stannylzinc chloride (nBu)₃SnZnCl•LiCl. The reaction setup is simple, using only cheap and commercially available copper(II) dibromide as precatalyst and bipyridine (bpy) as the ligand. Moreover, the cross-coupling proceeds with high chemo- and stereoselectivity. Thus, C(sp²)-O bond activation rather C(sp²)-Hal bond activation is observed, and starting from (Z)-alkenyl acetates, stereoretentive formation of the corresponding (Z)-alkenyl silanes and stannanes is obtained exclusively. The high functional group tolerance of this protocol also allows for late-stage functionalization. These are clearly the strong aspects of the present work that, in my opinion, would warrant publication in a top-tier journal such as *Nature Communications*.

However, there are also some drawbacks of the new method. In order to achieve the excellent stereospecificity ((Z)-alkenyl acetates  (Z)-alkenyl silanes/stannanes), a carboxyl directing-group is crucial (i.e., ester, lacton, or amide). Moreover, only tetrasubstituted alkenyl acetates undergo the silylation process, while trisubstituted alkenyl acetates lead to the hydrodesilylated products (i.e., compound 34). This significantly limits the substrate scope.

The experimental work has been carried out carefully (including an X-ray of cross-coupling product 109 to secure its conformation), and all results are adequately documented in the Supplementary Information. The proposed catalytic cycle is supported by mechanistic studies (including radical-clock and competition experiments) as well as DFT calculations, revealing oxidative addition of an in-situ generated cobalt(I) species into the C(sp²)-O bond of the alkenyl acetate and a subsequent rate-determining transmetalation as the key steps.

Overall, this is a nice piece of work, and I congratulate the authors for disclosing these new transformations. However, in its present form, I cannot recommend publication of this manuscript in *Nature Communications*. The manuscript simply contains too many technical errors, and the mechanistic picture contains inconsistencies (see below). The manuscript also needs careful proofreading (as do the Supplementary Information).

The following major concerns need to be addressed prior to publication:

1) Abstract: I would not say that this transformation was developed on the basis of an "Umpolung" strategy. Of course, the silyl- and stannylzinc reagents are inherently nucleophilic silicon and tin sources, while the polarity changes after C(sp²)-E (E = SiR₃ or ZnR₃) bond formation due to the different electronegativities. Otherwise, all C-E bond formations with silyl- and stannylzinc reagents

would have to be called “Umpolung” reactions.

2) Figure 1c (and the corresponding text) needs revision. This Figure is oversimplified and misleading. Left reaction path: The C–[M] reagent first comes into play in the transmetalation step after oxidative addition of the transition metal [TM] into the C(sp²)–OAc bond. Transmetalation is then followed by reductive elimination. This sequence becomes not clear enough. Right reaction path: The carbometallation is preceded by a transmetalation. At least, this should be mentioned to avoid confusion with the C–[M] reagent. Process II → III: This operation, labeled as “isomerization”, is an internal C–C bond rotation that is required for the subsequent syn-selective beta-OR elimination. For me, it is not clear, how the left product isomer is formed from intermediate II. Is LiCl playing a role in these transformations? Can the authors comment on this?

3) Page 10, lines 222 ff: There is no “carbocobaltation” here. Please revise!

4) The authors use a different model substrate for the C–O stannylation than for the C–O silylation. I assume this is based on the volatility and the toxicity of the tetraorganotin compounds. The authors should state on this and also give a safety advice in the Supplementary Information.

5) Page 15, Methods: The given general procedure in the manuscript is too general. The authors might want to specify the order of addition and the cooling source used.

Reviewer #3 (Remarks to the Author):

In this experimental work, Li et al reported on cobalt-catalyzed stereo-selective C-Si and C-Sn cross-coupling reactions between alkenyl acetates with an ester directing group and organozinc reagents to synthesize tri- and tetrasubstituted alkenyl silanes and stannanes, along with combined experimental and DFT mechanistic study on such interesting reactions. Similar transition-metal-catalyzed stereo-selective C-C coupling reactions between alkenyl acetates and organo-magnesium were literature-known (see: *J. Am. Chem. Soc.* 2019, 141, 18127–18135) with the same proposed mechanism based on oxidative insertion of C–O bond and ester directing group.

While the DFT calculations seemed to be consistent with the experimental mechanistic proposal, the quality of such calculations was not convincing for several reasons and thus could not really support the proposal. First, the over-simplified molecular models (no coordinating THF or CH₃CN, no stabilizing salt LiCl, assumed Co oxidation state, etc) could not even represent the role of OPiv-coordination on the unique reactivity of Me₂PhSi–ZnOPiv•Li₂(Cl)(OPiv) and (nBu)₃Sn–ZnCl•LiCl reagents observed in experiment. Why is OPiv-coordination crucial for C-Si but unfavorable for C-Sn coupling, and why should the reaction be kept on triplet potential energy surface? Second, the Zn-to-Co transfer of the Sn-group was proposed as the rate-limiting step involving three molecules (alkenyl acetate, Co catalyst, and Sn-Zn reagent) over a moderate free energy barrier of 21 kcal/mol; this may not be convincing for facile low-temperature reactions. Why is low-temperature so important for C-Sn coupling in experiment, and could the suggested kinetics be checked in experiment? Third, since ionic species were proposed especially for the rate-limiting step, gas-phase optimization for such TS and minima structures can be unreliable. Finally, the proposed key TS2 does not look like a true transition state, with the Co-Sn (3.46 Å) and Zn-Sn (3.25) distances being much too long than the computed Co-Sn (2.65) and Zn-Sn (2.56) single bonds; similarly in TS3, the C-Sn (2.25) distance is already the usual C-Sn (2.18) single bond along with a quite long Co-Sn (3.08) distance; the computed reaction path should be verified. Overall, the mechanistic proposal was not really supported by DFT calculations, making the paper weak for Nature Communications.

Reviewer #1 (Remarks to the Author):

Comments:

The submitted manuscript from Qi, Li, and co-workers focuses on a cobalt-catalyzed method for the substitution of vinyl acetate groups with silyl and stannyl nucleophiles. This results in the formation of new vinyl silanes and stannanes in a stereoselective manner, where the alkene stereochemistry is retained between starting material and product. The catalyst system - CoBr₂ and bipyridine - is simple, inexpensive, and readily available. The main group nucleophiles are somewhat more laborious to prepare, requiring formation of the lithium reagents from the silyl/stannyl chlorides, followed by transmetalation with ZnX₂. The reaction scope presented is very impressive, and products yields are synthetically useful. The products obtained (i.e. vinyl silanes/stannanes) are not themselves directly useful, but rather have potential as versatile intermediates for further functionalization/elaboration route to a more complex target. The authors demonstrate this point with some examples of further reactivity in Figure 8, mostly involving Stille coupling of the vinyl stannane derivatives.

Overall, this is a highly significant advance in synthetic methodology. Achieving selective activation/substitution of vinyl acetate derivatives is very challenging, due both to the strength of the C-O bond, and the myriad side reactions these substrates are prone to, such as competitive nucleophilic attack at the carbonyl carbon of the acetate group. Moreover, acyclic vinyl acetates provide an additional challenge in controlling the stereochemical outcome. While vinyl silane/stannane synthesis is perhaps less valuable than a direct substitution with carbon-based nucleophiles, the possibility of further reactivity

means this is a potentially useful approach. The demonstrated reaction scope and simplicity of the catalyst system are particular strengths of this work.

Response: We thank the reviewer for the positive comments.

This being said, there are several aspects of this work that should be addressed before publication is considered:

Q1) The new compounds all appear to be characterized by 1D NMR and MS methods (and in one specific case, XRD), though other methods determining composition and/or purity are not used (e.g. IR spectroscopy, melting points for solids, HPLC or GC traces for purity). While these alternative techniques are probably not necessary, there is the question of how the authors confirmed the product alkene stereochemistry for the acyclic products. There does not appear to be any mention/discussion of how this was accomplished (e.g. by NOE NMR techniques), nor any primary data in the SI addressing this.

Response: Thanks for these suggestions. We randomly selected nineteen products (11, 12, 18, 20, 23, 31, 42, 46, 50, 51, 52, 64, 71, 84, 84', 96, 106, 107) and measured their 2d NMR spectra for confirming the alkene stereochemistry, these results were added in the revised supporting information. Beyond that, we also randomly tested the purity of the isolated products 18, 20 (analysis by HPLC) and 46, 52, 64, 86, 86' (analysis by GC), the purity is fine and only single peak of the product can be found. Combination with the results of NMR, we are convinced of the purity of the resulting products. We have added the 2d-NMR spectra in the revised SI.

Q2) For the "mechanistic studies" in Figure 5, I'm not sure of the value of the "radical clock" experiment in 5a. Since the silylation and stannylation of this (E)-vinyl acetate do not actually seem to occur, then no conclusions can be made about 1-electron versus 2-electron pathways. Can the authors devise a more conclusive experiment to probe for radical intermediates?

Response: Thanks for the suggestions. To this end, a series of control experiments with stoichiometric amount of radical scavengers **82** such as BHT and 1,1-diphenylethene were performed, both silylated and stannylated products were isolated with a slightly reduced yields under the described conditions. These results indicated that the reaction may not involve a radical pathway. Additionally, *we have added these results in the revised manuscript and SI.*

(a) radical-clock experiment with radical scavengers **82**

Q3) To further probe the hypothesized chelation assisted oxidative addition, can the authors access a substrate where the ester group has been replaced by a nitrile? This would give a similarly resonance-activated vinyl acetate, but lack the key carbonyl oxygen for coordination to the Co center. The differential

reactivity between (Z) and (E) substrates observed (e.g. in Fig 5b and 5c) could be due to electronic differences rather than chelation assistance. Can the authors also computationally evaluate the predicted/calculated electrophilicity of (Z)-83 versus (E)-83? Perhaps the (Z) isomer is inherently more electrophilic, and that chelation assistance is not required to explain the reactivity.

Response: Herein, we further examined the effect of different directing groups by using the substrates (Z)-91, and 94. The substrate bearing a conjugated *N,N*-dimethyl amide (Z)-91 gave the desired stannane 93 in 43% yield with complete control of geometry. Moreover, an enol acetate 94 bearing a conjugated nitrile was also examined under the standard conditions. However, only trace amount of silylated product 95 and low yield of stannylated product 96 were formed. As expected, no desired C–O bond functionalization was observed when using (E)-91 or the substrate without a conjugated ester group as the electrophilic coupling partner. These findings suggested that the ester group is crucial to the stereoselective C–O bond activation due to its chelation assistance and conjugation effect to active the α,β -unsaturated alkenyl acetate system. Additionally, we have added these results in the revised manuscript and SI.

(e) effect of directing groups

Q4) For the calculated reaction pathway, did the authors similarly evaluate other potential pathways computationally? For example, an oxidative addition based mechanism where there is no chelation assistance, or the alkene insertion/ β -acetoxo elimination pathway? Also, please note that I am by no means a computational chemistry expert, and so I cannot comment on the computational methods used (another reviewer who is a computational expert should be consulted for this).

Response: Thanks for the reviewer's comments. Based on these suggestions, additional DFT calculations were performed to study the oxidative addition mechanism without chelation effect and the alkene insertion / β -acetoxo elimination mechanism. As shown below, an alkene-coordination intermediate ³98 was located before the C–O oxidative addition. Although this intermediate has a lower relative free energy than that of ester-coordination intermediate ³98b, the following C–O oxidative addition transition state ³TS-1 also contains the chelation of the ester group. The oxidative addition

transition state without chelation assistance is not located in this study. For the second potential reaction pathway, an alkene insertion transition state ${}^3\text{TS-2c}$ was located followed by the transmetalation of triplet Co(I) complex ${}^3\mathbf{97}$ with stannyl zinc reagent $\mathbf{100}$ [$({}^n\text{Bu})_3\text{Sn-ZnCl}\cdot\text{LiCl}\cdot\text{CH}_3\text{CN}$]. However, the activation free energy of ${}^3\text{TS-2c}$ ($\Delta G^\ddagger = 44.6$ kcal/mol) is very high, which indicates that the alkene insertion pathway is disfavored. These DFT results have been added to the **SI**.

Figure R1. DFT study of the oxidative addition mechanism and the alkene insertion /beta-acetoxy elimination mechanism. All energies were calculated at (U)B3LYP-D3/6-311+G(d,p)–SDD/SMD(acetonitrile)//(U)B3LYP-D3/6-31G(d)–LANL2DZ level of theory.

Q5) One additional paper that should be cited in the context of vinyl acetate substitution using main-group reagents is: New J. Chem., 2021, 45, 20095-20098 (borylation of vinyl acetates and pivalates using Pd catalysis, albeit with much more limited scope).

Response: we have appropriately cited the mentioned literature in the revised manuscript (ref. 37).

Q6) Finally, the manuscript text requires thorough editing/revision for grammar and diction. I will not itemize the needed changes here, as there are many (I count at least 10 in the first paragraph alone). As-written, the text errors are a significant distraction from the work itself. Given the high quality of the synthetic method, the authors no doubt would want a high quality text presentation as well.

Response: Thanks for the comments, we have carefully corrected the above mentioned issues in the revised manuscript

In summary, this work could certainly become suitable for publication in Nature Communications, provided the details on stereochemical confirmation are included, some additional clarity on the mechanism is provided, and the text itself is improved.

Reviewer #2 (Remarks to the Author):

Comments:

Qi, Li and co-workers describe cobalt-catalyzed cross-couplings of alkenyl acetates with silyl- and stannylzinc reagents to provide access to synthetically valuable polysubstituted alkenyl silanes and stannanes, respectively. This work builds on an earlier report of one of the authors in which the cross-coupling of alkenyl acetates with aryl or alkenyl zinc pivalates to form C(sp²)-C(sp²) bonds was realized under identical reaction conditions (see Ref. 25). Key to success for the unprecedented silylation and stannylation reactions is the use of the solid silylzinc pivaloate Me₂PhSiZnOPiv•Li₂(Cl)(OPiv) and stannylzinc chloride (nBu)₃SnZnCl•LiCl. The reaction setup is simple, using only cheap and commercially available copper(II) dibromide as precatalyst and bipyridine (bpy) as the ligand. Moreover, the cross-coupling proceeds with high chemo- and stereoselectivity. Thus, C(sp²)-O bond activation rather than C(sp²)-Hal bond activation is observed, and starting from (Z)-alkenyl acetates, stereoretentive formation of the corresponding (Z)-alkenyl silanes and stannanes is obtained exclusively. The high functional group tolerance of this protocol also allows for late-stage functionalization. These are clearly the strong aspects of the present work that, in my opinion, would warrant publication in a top-tier journal such as Nature Communications.

However, there are also some drawbacks of the new method. In order to achieve the excellent stereospecificity ((Z)-alkenyl acetates  (Z)-alkenyl silanes/stannanes), a carboxyl directing-group is crucial (i.e., ester, lactone, or amide). Moreover, only tetrasubstituted alkenyl acetates undergo the silylation process, while trisubstituted alkenyl acetates lead to the hydrodesilylated products (i.e., compound 34). This significantly limits the substrate scope.

The experimental work has been carried out carefully (including an X-ray of cross-coupling product **109** to secure its conformation), and all results are adequately documented in the Supplementary Information. The proposed catalytic cycle is supported by mechanistic studies (including radical-clock and competition experiments) as well as DFT calculations, revealing oxidative addition of an in-situ generated cobalt(I) species into the C(sp²)-O bond of the alkenyl acetate and a subsequent rate-determining transmetalation as the key steps.

Overall, this is a nice piece of work, and I congratulate the authors for disclosing these new transformations. However, in its present form, I cannot recommend publication of this manuscript in Nature Communications. The manuscript simply contains too many technical errors, and the mechanistic picture contains inconsistencies (see below). The manuscript also needs careful proofreading (as do the Supplementary Information).

Response: We thank the reviewer for the positive comments.

The following major concerns need to be addressed prior to publication:

Q1) Abstract: I would not say that this transformation was developed on the basis of an “Umpolung” strategy. Of course, the silyl- and stannylzinc reagents are inherently nucleophilic silicon and tin sources, while the polarity changes after C(sp²)-E (E = SiR₃ or SnR₃) bond formation due to the different electronegativities. Otherwise, all C-E bond formations with silyl- and stannylzinc reagents would have to be called “Umpolung” reactions.

Response: Thanks for the suggestions, we have deleted the “umpolung strategy” in the revised abstract.

Q2) Figure 1c (and the corresponding text) needs revision. This Figure is oversimplified and misleading. Left reaction path: The C–[M] reagent first comes into play in the transmetalation step after oxidative addition of the transition metal [TM] into the C(sp²)–OAc bond. Transmetalation is then followed by reductive elimination. This sequence becomes not clear enough. Right reaction path: The carbometallation is preceded by a transmetalation. At least, this should be mentioned to avoid confusion with the C–[M] reagent. Process II  III: This operation, labeled as “isomerization”, is an internal C–C bond rotation that is required for the subsequent syn-selective beta-OR elimination. For me, it is not clear, how the left product isomer is formed from intermediate II. Is LiCl playing a role in these transformations? Can the authors comment on this?

Response: We have reorganized the Figure 1c in the revised manuscript. Initial transmetalation and oxidative insertion formed the single diastereomer intermediate (I), which undergoes stereoretentive cross-coupling process. For the right reaction pathway, an initial transmetalation between organometallics and transition metal forms the C–[TM] species, which undergoes 1,2-carbometallation of enol carboxylates, in tandem with isomerization (II and III) and *trans*-, or *syn*-elimination, thereby delivering the *Z/E*-mixtures. We have added the following sentence in the revised manuscript: “*Toward the mechanism insights of C–O bond activation, two plausible pathways have thus far acknowledged: initial transmetalation between organometallics and transition metal forms the C–[TM] species, which undergoes i) oxidative insertion of C–O bond to the [TM]-center, along with following reductive elimination; or ii) 1,2-carbometallation of enol carboxylates, in tandem with isomerization (II and III) and trans-, or syn-elimination.*”.

Q3) Page 10, lines 222 ff: There is no “carbocobaltation” here. Please revise!

Response: We have corrected the sentence in the revised manuscript: “*These findings suggested that the chelation-assisted oxidative insertion of cobalt into the C–O bond of (Z)-alkenyl acetates is preferred to (E)-alkenyl acetates.*”

Q4) The authors use a different model substrate for the C–O stannylation than for the C–O silylation. I assume this is based on the volatility and the toxicity of the tetraorganotin compounds. The authors should state on this and also give a safety advice in the Supplementary Information.

Response: Thanks for the kind advice, we have added a safety advice for the preparation of Sn–Zn reagents in the revised supporting information. To evaluate the substrate scope of polyfunctional alkenyl acetates, we randomly selected the electrophiles for the envisioned cobalt-catalyzed stannylation. As we can see, the product **38** was also obtained in 85% yield under our reaction conditions.

Q5) Page 15, Methods: The given general procedure in the manuscript is too general. The authors might want to specify the order of addition and the cooling source used.

Response: Thanks for the suggestions, we have rewritten the methods in the revised manuscript (see Page 15).

Reviewer #3 (Remarks to the Author):

Comments:

In this experimental work, Li et al reported on cobalt-catalyzed stereo-selective C-Si and C-Sn cross-coupling reactions between alkenyl acetates with an ester directing group and organozinc reagents to synthesize tri- and tetrasubstituted alkenyl silanes and stannanes, along with combined experimental and DFT mechanistic study on such interesting reactions. Similar transition-metal-catalyzed stereo-selective C-C coupling reactions between alkenyl acetates and organo-magnesium were literature-known (see: J. Am. Chem. Soc. 2019, 141, 18127–18135) with the same proposed mechanism based on oxidative insertion of C–O bond and ester directing group.

While the DFT calculations seemed to be consistent with the experimental mechanistic proposal, the quality of such calculations was not convincing for several reasons and thus could not really support the proposal.

Response: We thank the reviewer for the comments. Based on these constructive comments, additional DFT calculations were performed to improve the quality of computational studies in the revised manuscript. The structure of Sn-Zn and Si-Zn reagents in the solvent and the lithium salt effect are studied. The complete free energy profile with Si-Zn reagent is also calculated. The point-by-point responses are shown below:

Q1) First, the over-simplified molecular models (no coordinating THF or CH₃CN, no stabilizing salt LiCl, assumed Co oxidation state, etc) could not even represent the role of OPiv-coordination on the unique reactivity of Me₂PhSi–ZnOPiv•Li₂(Cl)(OPiv) and (nBu)₃Sn–ZnCl•LiCl reagents observed in experiment. Why is OPiv-coordination crucial for C-Si but unfavorable for C-Sn coupling?

Response: Thanks for these comments. In the revised manuscript, the real structures of Sn-Zn and Si-Zn reagents were carefully studied, including the coordination of solvent (THF or CH₃CN) and the effect of salt LiCl. As shown below (Figure R2), the coordination of one CH₃CN to **100** is endergonic by 9.7 kcal/mol and the dissociation of lithium salt **101** (LiCl•CH₃CN) from **100** is endergonic by 0.1 kcal/mol. These results indicate that the Sn-Zn reagent **100** is a more stable structure. The structure of intermediate **103** is also investigated. The computational results suggest that the coordination of lithium salt to Zn is thermodynamically disfavored.

Figure R2. DFT study of the most stable structure of Sn-Zn reagent. All energies were calculated at (U)B3LYP-D3/6-311+G(d,p)–SDD/SMD(acetonitrile)//(U)B3LYP-D3/6-31G(d)–LANL2DZ level of theory.

Although the Sn-Zn reagent **100** has a reasonable stability, its low reactivity in transmetalation process witnessed by the higher activation free energy (Figure R3, **³TS-2b**, $\Delta G^\ddagger = 33.6$ kcal/mol) indicates that lithium salt **101** will dissociate before transmetalation. The most favored reaction pathway is shown in Figure 6.

Figure R3. Free energy profile of cobalt(I)-catalyzed cross-coupling reaction of alkenyl acetate **83** with **100** ($(n\text{Bu})_3\text{Sn-ZnCl}\cdot\text{LiCl}\cdot\text{CH}_3\text{CN}$). The energies were calculated at (U)B3LYP-D3/6-311+G(d,p)-SDD/SMD(acetonitrile)/(U)B3LYP-D3/6-31G(d)-LANL2DZ level of theory. Bond lengths are given in angstrom (Å).

As shown in Figure R4, the most stable structure of Si-Zn reagent is **120**. The coordination of THF to **120** is endergonic by 7.5 kcal/mol and the dissociation of $\text{LiCl}\cdot\text{THF}$ is endergonic by 10.2 kcal/mol. The Si-Zn reagent **120** involved reaction pathway has also been studied and the complete free energy profile was summarized in Figure S8. The computational results leads to the similar conclusion as that of Sn-Zn reagent involved cross-coupling reaction. In addition, the well-defined Co(0) source, such as $\text{Co}_2(\text{CO})_8$ and $\text{Co}(\text{phen}^{\text{Pr}})(\text{dvtms})$ (prepared according to the reference: *Angew. Chem. Int. Ed.* **2019**, *58*, 1552-1556), failed to deliver the desired products.

Figure R4. DFT study of the most stable structure of Si-Zn reagent. All energies were calculated at (U)B3LYP-D3/6-311+G(d,p)–SDD/SMD(acetonitrile)//(U)B3LYP-D3/6-31G(d)–LANL2DZ level of theory.

Moreover, Lei and co-workers demonstrated that arylzinc reagents prepared by different methods possess very different kinetics in palladium- and nickel-catalyzed oxidative couplings, and further X-ray absorption spectroscopy studies show that changing the halide anion from Cl to Br or I will result in an increase of the Zn–C bond distance and thereby improve the transmetalation rate (*J. Am. Chem. Soc.* 2009, **131**, 16656-16657; *Chem. Commun.* 2013, **49**, 9615-9617; *Chem. Commun.* 2014, **50**, 8709-8711). Hence, we proposed that the anion-regulation of Si-Zn, or Sn-Zn reagents might affect these cobalt-catalyzed cross-coupling reactions.

Q2) Why should the reaction be kept on triplet potential energy surface?

Response: In this study, both singlet and triplet free energy profiles have been considered in DFT calculation. Computational studies suggest that the triplet CoBr(bpy) **397** is more stable than the singlet complex by 29.7 kcal/mol (Figure R5). Subsequent C–O oxidative addition to cobalt(I) (via **1TS-1**) on singlet free energy profile also requires a high barrier (**1TS-1** vs **3TS-1**, $\Delta G^\ddagger = 31.0$ vs 13.7 kcal/mol, respectively). These results suggest that the cobalt(I)-catalyzed cross-coupling reaction prefers to occur on triplet potential energy surface. This conclusion is consistent with previous computational studies regarding Co(I) catalysis, see refs.: Qi, X. Liu, X. Qu, L.-B. Liu, Q. Lan, Y. Mechanistic insight into cobalt-catalyzed stereodivergent semihydrogenation of alkynes: The story of selectivity control. *J. Catal.* 362, 25–34 (2018); Also see a review for cobalt-catalyzed cross-couplings: Capdevila, L. Ribas, X. PATAI'S Chemistry of Functional Groups: Cobalt-catalyzed cross-coupling reactions. (Wiley, 2022).

Figure R5. All energies were calculated at (U)B3LYP-D3/6-311+G(d,p)–SDD/SMD(acetonitrile)//(U)B3LYP-D3/6-31G(d)–LANL2DZ level of theory. All energies are with respect to triplet Co(I) complex **397**. Bond lengths are given in angstrom (Å).

Q3) Second, the Zn-to-Co transfer of the Sn-group was proposed as the rate-limiting step involving three molecules (alkenyl acetate, Co catalyst, and Sn-Zn reagent) over a moderate free energy barrier of 21

kcal/mol; this may not be convincing for facile low-temperature reactions. Why is low-temperature so important for C-Sn coupling in experiment, and could the suggested kinetics be checked in experiment?

Response: Generally, the trisubstituted alkenyl acetates are more reactive than the tetrasubstituted substrates. Cobalt-catalyzed stannylation with trisubstituted acyclic acetate only gave the desired product in low yields when performing the reaction under 0 or 23 °C. Significant amount of the corresponding hydrodestannylated products and ketoesters were formed, which is likely owing to the poor stability of the alkenyl stannanes and the strong alkalinity of the Sn-Zn reagent under the reaction conditions. Nevertheless, the stannylation and silylation of tetrasubstituted alkenyl acetates were occurred under 0 to 23 °C.

Q4) Third, since ionic species were proposed especially for the rate-limiting step, gas-phase optimization for such TS and minima structures can be unreliable.

Response: To compare the solvent effect on geometry optimization, the C-O oxidative addition was recalculated in the solvent. The reactant, intermediates, and transition state were reoptimized in acetonitrile. The comparison between gas-phase optimization and solvent optimization suggest that the solvent effect is negligible on C-O oxidative addition step. We also tried to reoptimize the transmetalation transition state in acetonitrile, however, the geometry optimization cannot be accomplished after many attempts. We then recalculate the formation of pentacoordinated cobalt(III) complex **3104** from intermediate **399**. As shown in Figure R6, the energy difference between gas-phase optimization and solvent optimization is only 3.2 kcal/mol. Based on these results, we surmise that the geometry optimization in gas-phase is reliable for this reaction system.

Figure R6. Comparison of the DFT results obtained by geometry optimization in gas-phase and solvent. The red numbers represent the energies calculated at (U)B3LYP-D3/6-311+G(d,p)-SDD/SMD(acetonitrile)//(U)B3LYP-D3/6-31G(d)-LANL2DZ(acetonitrile) level of theory. All energies are with respect to triplet Co(I) complex **397**.

Q5 Finally, the proposed key TS2 does not look like a true transition state, with the Co-Sn (3.46 Å) and Zn-Sn (3.25) distances being much too long than the computed Co-Sn (2.65) and Zn-Sn (2.56) single bonds; similarly in TS3, the C-Sn (2.25) distance is already the usual C-Sn (2.18) single bond along with a quite long Co-Sn (3.08) distance; the computed reaction path should be verified.

Response: Thanks for the comments. In our DFT calculation, intrinsic reaction coordinate (IRC) has been performed to verify the reliability of the transition states. The IRC calculation results of **³TS-2** and **³TS-3** have been summarized:

a) IRC calculation of transition state ³TS-2

IRC calculation results of transmetalation transition state **³TS-2** are illustrated in Figure R7. The structural information of selected points on IRC profile are also given. These results show that the Co-Sn bond decreases from 4.45 Å (structure **d**) to 3.46 Å (**³TS-2**) and then gradually decreases to 2.86 Å (structure **C**). The variation of Co-Sn distance verifies that a covalent Co-Sn bond is formed through transition state **³TS-2**, which leads to the formation of intermediate **105**.

Figure R7. Intrinsic reaction coordinate (IRC) calculation of transition state **TS-2**. The structural information of selected points on IRC profile is given. Bond lengths are given in angstrom.

b) IRC calculation of transition state TS-3

IRC calculation results of C-Sn reductive elimination transition state **TS-3** are illustrated in Figure R8. The structural information of selected points on IRC profile are also given. These results show that the Co-Sn bond decreases from 3.42 Å (structure **d**) to 3.08 Å (**TS-3**) and then gradually decreases to 2.71 Å (structure **C**). The variation of Co-Sn distance verifies that a covalent C-Sn bond is formed through transition state **TS-3**, which leads to the formation of product **84**.

Figure R8. Intrinsic reaction coordinate (IRC) calculation of transition state **TS-3**. The structural information of selected points on IRC profile is given. Bond lengths are given in angstrom.

REVIEWER COMMENTS

Reviewer #1 (Remarks to the Author):

The revised manuscript from Qi, Ji, and co-workers has addressed my specific comments for revisions. As presented now, the experimental work is suitable for acceptance. I defer to Reviewer 3 to determine if the revisions to the computational work are sufficient.

Minor additional text/figure comments:

- Figure 1: throughout: precedents -> precedent; c) tans -> trans, also remove "isomerization" from the equilibrium arrow (as Reviewer 2 points out, as drawn this is a bond rotation, not an isomerization); d) remove "Umpolung" from descriptor (to be consistent with Reviewer 2's prior comments)
- Pg 3, line 78: acohols -> alcohols
- Pg 3, line 83: remove "via umpolung strategy" (to be consistent with Reviewer 2's prior comments)
- Pg 3, line 85: aommonly -> commonly
- Pg 4, line 103: Furthermore. -> Furthermore,
- Figure 5(a): these are not "radical clock" experiments (same in the SI, page S-5)
- Pg 11, line 242: describing a reaction step as "tricky" is too colloquial
- Figure 8: The "tert" nomenclature is incorrect, and this figure also represents reactions of both alkenyl silanes and stannanes - the additional figure title at the top is likely unnecessary anyway, since there is a descriptive figure caption below.

Reviewer #2 (Remarks to the Author):

In their revised manuscript, Qi, Li and co-workers have addressed all comments and requests of all three reviewers without exception point by point. The work now includes

- additional information on the product purity (HPLC or GC traces) and assignment of the alkene stereochemistry by 2D NMR spectra (see the SI)
- further mechanistic studies such as control experiments with radical scavengers (see Scheme 5a) and the (unsuccessful) use of nitrile as directing group (see Scheme 5e)

I have no further concerns regarding the experimental part of this study.

As a major revision, the authors performed additional DFT calculations to address the comments raised by Reviewer 1 and, in particular, Reviewer 3. These calculations (see new Figures S1–S9 in the revised SI) include

- an oxidative addition mechanism without chelation effect and alkene insertion/beta-acetoxy elimination mechanism (see Figure S1)
- the structures of the Sn–Zn and Si–Zn reagents with coordination of solvent and LiCl (see Figures S3 and S4) and the effect on the transmetalation step (see Figure S5)
- a comparison of the oxidative addition by geometry between optimization in the gas-phase and solvent (see Figure R6, Q4 by Reviewer 3)
- intrinsic reaction coordination calculation of transition state 3TS-2 and 3TS-3 (see Figures S6 and S7)

Since I am not an expert in computational chemistry, I cannot comment on the computational methods and results. Another reviewer should be consulted to evaluate these new data.

Minor things:

- 1) Figure 1c, left pathway: Isn't it first oxidative addition and then transmetalation, and shouldn't the OAc group be removed from intermediate I?
- 2) Figure 5a: The use of a radical scavenger is not a "radical-clock experiment". Please change

“radical-clock experiment” by “control experiment”.

3) In my opinion, the title of the manuscript is too general and implies a review rather than an original article.

Reviewer #3 (Remarks to the Author):

In this revision, single-point B3LPY-D3/SDD/6-311+G(d,p) + SMD(acetonitrile) calculations based on B3LYP-D3/LanL2DZ/6-31G(d) gas-phase geometries are used to explore the catalytic reaction mechanism in acetonitrile solution. As pointed out in last review comments, the quality of such calculations using gas-phase geometries and over-simplified model system is not convincing to support the mechanistic proposal. However, such problem remains mostly unchanged in this revision. First, the assumed tri-coordinated Co(I) catalyst CoBpyBr is questionable starting from CoBr₂ salt and pyridine ligands. Test DFT calculations show that CH₃CN molecule is about 4 kcal/mol bound to triplet CoBpyBr, which will evidently increase the overall barrier. The fact that triplet CoBpyBr is more stable than its singlet state does not mean the whole catalytic reaction should be kept on triplet state with additional substrates. Changing multiplicity of transition metal complex during catalysis is rather common. Second, taking solvent-coordination to CoBpyBr into account, the Zn-to-Co transfer of the Sn-group was proposed as the rate-limiting step involving three molecules (alkenyl acetate, Co-catalyst, and Sn-Zn reagent) over a sizeable free energy barrier of 26 kcal/mol, which cannot account for the facile low-temperature C–Sn coupling observed in experiment. Third, ionic species were proposed especially for the rate-limiting step, gas-phase optimization for such TS and minima structures can be unreliable, with the proposed key TS₂ and TS₃ are very likely due to artifact of gas-phase calculations rather than true TS in solution. In fact, the authors cannot even reproduce such TSs when solvation model is included, while in a tested example of ionic reaction the energy difference reached already 3.2 kcal/mol. Geometry optimization in solution is thus indeed important to get reliable energetics. Overall, the mechanistic proposal was not really supported by the present DFT calculations using gas-phase geometries and over-simplified model system. The authors should take these points seriously to solve the mechanistic problem before the manuscript could be accepted for publication in Nature Communications.

Reviewer #1 (Remarks to the Author):

Comments:

The revised manuscript from Qi, Li, and co-workers has addressed my specific comments for revisions. As presented now, the experimental work is suitable for acceptance.

Response: We thank the reviewer for the positive comments.

Q1) Minor additional text/figure comments:

-Figure 1: throughout: precedents -> precedent; c) tans -> trans, also remove "isomerization" from the equilibrium arrow (as Reviewer 2 points out, as drawn this is a bond rotation, not an isomerization); d) remove "Umpolung" from descriptor (to be consistent with Reviewer 2's prior comments)

-Pg 3, line 78: acohols -> alcohols

-Pg 3, line 83: remove "via umpolung strategy" (to be consistent with Reviewer 2's prior comments)

-Pg 3, line 85: aommonly -> commonly

-Pg 4, line 103: Furthermore. -> Furthermore,

-Figure 5(a): these are not "radical clock" experiments (same in the SI, page S-5)

-Pg 11, line 242: describing a reaction step as "tricky" is too colloquial

-Figure 8: The "tert" nomenclature is incorrect, and this figure also represents reactions of both alkenyl silanes and stannanes - the additional figure title at the top is likely unnecessary anyway, since there is a descriptive figure caption below.

Response: The above mentioned errors have been corrected in the revised manuscript.

Reviewer #2 (Remarks to the Author):

Comments:

In their revised manuscript, Qi, Li and co-workers have addressed all comments and requests of all three reviewers without exception point by point. The work now includes

- additional information on the product purity (HPLC or GC traces) and assignment of the alkene stereochemistry by 2D NMR spectra (see the SI)

- further mechanistic studies such as control experiments with radical scavengers (see Scheme 5a) and the (unsuccessful) use of nitrile as directing group (see Scheme 5e)

I have no further concerns regarding the experimental part of this study..

Response: We thank the reviewer for the positive comments.

Q1) Minor things:

1) Figure 1c, left pathway: Isn't it first oxidative addition and then transmetalation, and shouldn't the OAc group be removed from intermediate I?

Response: Thanks for the suggestions. We have revised the intermediate I.

2) Figure 5a: The use of a radical scavenger is not a "radical-clock experiment". Please change "radical-clock experiment" by "control experiment".

Response: We have corrected this error in the revised manuscript.

3) In my opinion, the title of the manuscript is too general and implies a review rather than an original article.

Response: Thanks for the suggestions, we have revised the title to "**Stereoselective C–O Silylation and Stannylation of Alkenyl Acetates**".

Reviewer #3 (Remarks to the Author):

Comments:

Q1) In this revision, single-point B3LPY-D3/SDD/6-311+G(d,p) + SMD(acetonitrile) calculations based on B3LYP-D3/LanI2DZ/6-31G(d) gas-phase geometries are used to explore the catalytic reaction mechanism in acetonitrile solution. As pointed out in last review comments, the quality of such calculations using gas-phase geometries and over-simplified model system is not convincing to support the mechanistic proposal. However, such problem remains mostly unchanged in this revision. First, the assumed tri-coordinated Co(I) catalyst CoBpyBr is questionable starting from CoBr₂ salt and pyridine ligands. Test DFT calculations show that CH₃CN molecule is about 4 kcal/mol bound to triplet CoBpyBr, which will evidently increase the overall barrier. The fact that triplet CoBpyBr is more stable than its singlet state does not mean the whole catalytic reaction should be kept on triplet state with additional substrates. Changing multiplicity of transition metal complex during catalysis is rather common.

Response: Thanks for the reviewer's comments. These comments focus on the employed computational methods, the spin state and the structures of cobalt intermediates. Based on the reviewer's comments, the hybrid GGA meta-functional, M06 has been used for single-point energy calculation in the revised manuscript. Thanks to the reviewer's suggestion, the coordination of the solvent (CH₃CN or THF) to Co(I) complex CoBpyBr was studied. As shown below, the coordination of one acetonitrile to CoBpyBr is exergonic by 2.9 kcal/mol, which is consistent with the reviewer's test calculation. The coordination of THF to CoBpyBr is endergonic. Therefore, the starting point of the Sn-Zn reagent involved free energy profile has been changed and the numberings are also revised.

For the spin state of cobalt complexes, in our DFT studies, both triplet and singlet states have been considered for all the intermediates, including Co(I) and Co(III) intermediates. As shown below, computational results suggest that the triplet state is favored for all the intermediates in this work. This conclusion is also consistent with the previous theoretical study of cobalt catalysis (Liu et al. J. Am. Chem. Soc. 2018, 140, 6873).

Q2) Second, taking solvent-coordination to CoBpyBr into account, the Zn-to-Co transfer of the Sn-group was proposed as the rate-limiting step involving three molecules (alkenyl acetate, Co-catalyst, and Sn-Zn reagent) over a sizeable free energy barrier of 26 kcal/mol, which cannot account for the facile low-temperature C-Sn coupling observed in experiment.

Response: Thanks for the reviewer's comments. In the updated free energy profile, M06 calculated activation free energy of Sn-Zn reagent involved transmetalation is 26.9 kcal/mol, while B3LYP-D3 calculated activation free energy is 24.2 kcal/mol. We are aware that this energy barrier seems like higher for this reaction as the reaction temperature is only $-20\text{ }^{\circ}\text{C}$. We have tried to locate more stable conformations for the transmetallation but failed. We also tried to reoptimize the transmetalation transition state in the solvent, however, the structures always collapse during the geometry optimization.

Q3) Third, ionic species were proposed especially for the rate-limiting step, gas-phase optimization for such TS and minima structures can be unreliable, with the proposed key TS2 and TS3 are very likely due to artifact of gas-phase calculations rather than true TS in solution. In fact, the authors cannot even reproduce such TSs when solvation model is included, while in a tested example of ionic reaction the energy difference reached already 3.2 kcal/mol. Geometry optimization in solution is thus indeed important to get reliable energetics.

Response: Thanks for the reviewer's comments. We totally agree with the reviewer's opinion that geometry optimization in solvent is significant for energy calculation, especially for computational study of ionic reactions. In this study, many efforts have been devoted to reoptimizing the structures in solvent. Now we have obtained the structures of intermediates and transition states in C–O oxidative addition and C–Sn reductive elimination steps. The activation free energy of C–O oxidative addition transition state optimized in solvent is close to that obtained in gas-phase (13.2 vs 13.7 kcal/mol). Meanwhile, the activation free energy of C–Sn reductive elimination transition state optimized in solvent is also close to that obtained in gas-phase (5.7 vs 5.4 kcal/mol). Unfortunately, the transmetalation transition state is still not obtained after many attempts. We would like to defend for ourselves that the geometry optimization in solvent is actually very challenging as the use of implicit solvation model will render the geometry convergence more difficult. We reckon the reviewer must be strict and prudent at research and we really appreciate the useful comments. Meanwhile, we sincerely hope that he/she can understand our situation. If the reviewer is still unsatisfied with the present DFT results, we can remove all the DFT results to the SI or even delete all the DFT results from this work.